# FBXO7/ntc and USP30 antagonistically set the ubiquitination threshold for basal mitophagy and provide a target for Pink1 phosphorylation in vivo

**Alvaro Sanchez-Martinez**◉*, **Aitor Martinez**◉, **Alexander J. Whitworth**◉*

MRC Mitochondrial Biology Unit, University of Cambridge, Cambridge Biomedical Campus, Cambridge, United Kingdom

* alvaro.sanchez@mrc-mbu.cam.ac.uk (ASM); a.whitworth@mrc-mbu.cam.ac.uk (AJW)

**Data Availability Statement:** All data needed to evaluate the conclusions in the paper are present in the paper and/or the Supplementary Materials. This

## Abstract

Functional analyses of genes linked to heritable forms of Parkinson's disease (PD) have revealed fundamental insights into the biological processes underpinning pathogenic mechanisms. Mutations in *PARK15/FBXO7* cause autosomal recessive PD and FBXO7 has been shown to regulate mitochondrial homeostasis. We investigated the extent to which FBXO7 and its *Drosophila* orthologue, ntc, share functional homology and explored its role in mitophagy in vivo. We show that *ntc* mutants partially phenocopy *Pink1* and *parkin* mutants and *ntc* overexpression supresses *parkin* phenotypes. Furthermore, ntc can modulate basal mitophagy in a Pink1- and parkin-independent manner by promoting the ubiquitination of mitochondrial proteins, a mechanism that is opposed by the deubiquitinase USP30. This basal ubiquitination serves as the substrate for Pink1-mediated phosphorylation that triggers stress-induced mitophagy. We propose that FBXO7/ntc works in equilibrium with USP30 to provide a checkpoint for mitochondrial quality control in basal conditions in vivo and presents a new avenue for therapeutic approaches.

## Introduction

Parkinson's disease (PD) is the second most common neurodegenerative disorder. Autosomal recessive mutations in the genes encoding the mitochondrial kinase PINK1 (*PINK1*) and the E3 ubiquitin ligase Parkin (*PRKN*) are associated with early-onset parkinsonism. These genes have been shown to function in a common mitochondrial quality control pathway whereby mitochondria are degraded by autophagy (mitophagy). Briefly, PINK1 is partially imported to healthy polarised mitochondria where it is cleaved and subsequently degraded in the cytosol [1]. Upon mitochondrial depolarisation, PINK1 is stabilised on the outer mitochondrial membrane (OMM) where it phosphorylates ubiquitin-Ser65 (pS65-Ub) conjugated to OMM proteins [2,3]. This acts as a signal for Parkin to be recruited and phosphorylated, relieving its autoinhibitory conformation, and allowing it to further ubiquitinate OMM proteins in close proximity [4,5] that will serve as additional substrates for PINK1 creating a feed-forward

study includes no data deposited in external repositories.

**Funding:** This work was supported by MRC core funding (MC_UU_00028/6 to A.J.W.) and by the Basque Government Postdoctoral Fellowship (POS_2022_2_0045 and salary to A.M.). The funders had no role in study design, data collection and analysis, decision to publish, or preparation of the manuscript.

**Competing interests:** The authors have declared that no competing interests exist.

**Abbreviations:** ADP, adenosine diphosphate; CNS, central nervous system; OMM, outer mitochondrial membrane; PD, Parkinson's disease; SEM, scanning electron microscopy; siRNA, small interfering RNA; TEM, transmission electron microscopy.

mechanism [6]. Counteracting mitochondrial ubiquitination, USP30, a mitochondrial outer membrane deubiquitinase, removes ubiquitin (Ub) from Parkin substrates acting as a negative regulator of mitophagy [7]. The accumulation of pS65-Ub on the OMM triggers the recruitment of autophagy receptors, which promote autophagosome recruitment and, ultimately, degradation of the damaged mitochondria [8,9].

However, most of our understanding of the function of PINK1 and Parkin come from the utilisation of mitochondrial uncouplers or inhibitors in cultured cells, usually in conjunction with Parkin overexpression [8,10]. Hence, there is a need to understand these mechanisms in more physiological model systems. *Drosophila melanogaster* models have delivered fundamental insights into the physiological function of the homologues *Pink1* and *parkin* (adopting FlyBase nomenclature), providing an excellent system to study mitochondrial homeostasis [11–16]. Interestingly, while proteomic analysis of mitochondrial turnover in *Drosophila* supports a role of Pink1/parkin in mitochondrial quality control [17], studies using pH-sensitive fluorescent mitophagy reporters showed that Pink1 and parkin had minimal impact on basal mitophagy [18–21]. Thus, many questions endure regarding the mechanisms of action of Pink1/parkin in mitochondrial quality control in vivo and the potential contribution of other key players.

Mutations in the gene encoding F-box protein 7 (*FBXO7*) have been found to cause autosomal recessive early-onset parkinsonism similar to those caused by mutations in *PINK1* and *PRKN* [22,23], motivating a need to understand how FBXO7 is related to PINK1/Parkin biology. F-box domain-containing proteins are essential factors of the SCF-type (Skp1-Cul1-F-box) E3-ubiquitin ligase complexes, which are responsible for recruiting Ub target substrates via the F-box domain [24]. FBXO7 has both SCF-dependent and SCF-independent activities [25,26]. Although it has been previously shown to cooperate with PINK1 and Parkin in mitophagy [27], how this occurs in vivo is poorly characterised warranting further investigation in an animal model. *Drosophila* encode a single homologue of *FBXO7* called *nutcracker* (*ntc*), which was identified in a screen for genes that control caspase activation during a late stage of spermatogenesis [28]. ntc shares sequence, structure, and some functional similarities with mammalian FBXO7 [26,28,29] and has been shown to regulate proteasome function via its binding partner PI31, an interaction that is conserved in FBXO7 [28,29]. Thus, *Drosophila* provides a genetically tractable system to study role of ntc/FBXO7 in mitochondrial quality control in vivo.

Mitochondrial quality control mechanisms are believed to have an important pathophysiological role in PD. However, how this process is orchestrated in vivo and the upstream factors involved are still unclear. Therefore, we sought to investigate the relationship between ntc, Pink1 and parkin, and their role in mitophagy, in particular basal mitophagy. We have found that ntc is able to compensate for the absence of *parkin* but not *Pink1*. Moreover, ntc plays an important role in promoting basal mitophagy in a Pink1- and parkin-independent manner, which is opposed by its counteracting partner USP30. This mechanism sets a threshold for basal mitophagy by regulating the steady-state levels of ubiquitin on OMM proteins, subsequently modulating the levels of pS65-Ub. Together, we provide evidence of a novel checkpoint for basal mitophagy that could potentially serve as therapeutic target for neurodegenerative disorders.

## Results

### Overexpression of ntc can rescue parkin but not Pink1 mutant phenotypes

The F-box domain localised in the C-terminal of *Drosophila* ntc shares considerable sequence similarity with that of human FBXO7 [28]. Both proteins have a conserved valine residue

involved in substrate recognition, including common interacting partners, e.g., PI31 [26,29]. However, a functional conservation in between FBXO7 and ntc has not been formally tested in vivo. Notably, it has been shown that FBXO7 functionally interacts with mammalian PINK1 and Parkin and genetically interacts with *parkin* in *Drosophila* [27]. Therefore, as an initial exploration of the functional homology between ntc and FBXO7, we tested for a similar genetic interaction between *ntc* and *parkin*. *Drosophila parkin* mutants present multiple disease-relevant phenotypes, including locomotor deficits, dopaminergic neurodegeneration, and a severe mitochondrial disruption leading to flight muscle degeneration [13,14]. Strikingly, overexpression of *ntc* by the strong ubiquitous *daughterless* (*da*)-GAL4 driver significantly rescued these *parkin* phenotypes (Fig 1A–1D). At the molecular level, we observed that ntc is also able to decrease the steady-state levels of the *Drosophila* Mitofusin (MFN1/MFN2) homologue, Marf, which has been previously shown to be increased in *parkin* mutants [16] (Fig 1E). These results support a functional homology between FBXO7 and ntc, and, moreover, indicate that ntc can at least partially substitute for parkin in vivo.

In contrast, assessing the functional relationship between ntc and Pink1 in a similar manner, *ntc* overexpression was unable to rescue climbing or flight locomotor defects of *Pink1* mutants, nor the increased Marf steady-state levels (Fig 1F–1H). Furthermore, the neurodegenerative "rough eye" phenotype—commonly used for testing genetic interactors [30,31]—induced by *Pink1* overexpression, in a robustly stereotyped manner, was not enhanced by *ntc* overexpression as previously shown for *parkin* [15]. These results mirror equivalent analyses of *FBXO7* [27] and indicate that *Pink1* and *ntc* do not obviously genetically interact.

## *ntc* mutants partially phenocopy *Pink1/parkin* mutants

If ntc performs similar functions to the Pink1/parkin pathway in *Drosophila*, we reasoned that *ntc* mutants may phenocopy *Pink1/parkin* mutants. Supporting this, *ntc* mutants, like *Pink1* and *parkin* mutants, are male sterile due to defective sperm individualization [11–13,28,32]. To extend this, we assessed *ntc* mutants for classic *Pink1/parkin* phenotypes as described above. *ntc* mutants homozygous for an amorphic nonsense mutation ($ntc^{ms771}$; abbreviated as $ntc^{-/-}$) showed a marked defect in climbing and flight ability (Fig 2A and 2B). We verified this was likely specific to *ntc* by assessing a hemizygous condition ($ntc^{-/Df}$) with comparable results (Fig 2A and 2B). Importantly, transgenic re-expression of *ntc* was able to significantly rescue the climbing and flight deficit (Fig 2A and 2B), confirming that the phenotype is specifically caused by loss of ntc function. Also similar to *Pink1/parkin* mutants, *ntc* mutants showed a dramatically shortened lifespan, with median survival of approximately 7 days compared to approximately 55 days for controls (Fig 2C), which was almost completely rescued by transgenic re-expression of *ntc* (Fig 2C).

However, in contrast to *Pink1/parkin* mutants, we did not observed loss of DA neurons in the PPL1 clusters of *ntc* mutants, although the short lifespan limited the analysis to only 10-day-old flies (Fig 2D). Likewise, immunohistochemical analyses of *ntc* mutants revealed no apparent disruption of mitochondrial morphology or integrity at confocal or electron-microscopy levels in flight muscle (Fig 2E) or larval central nervous system (CNS) (Fig 2F). Analysing mitochondrial respiratory capacity, *ntc* mutants were not consistently different from controls despite a downward trend (Fig 2G). Nevertheless, ATP levels were reduced in *ntc* mutants (Fig 2H), which was rescued by re-expression of wild-type *ntc*.

Oxidative stress is a consistent feature of PD pathology and normal ageing. Many models of PD show sensitivity to oxidative stressors such as paraquat (PQ), including *Drosophila parkin* mutants [33,34]. Similarly, *ntc* mutants also showed increased sensitivity to PQ exposure (Fig 2I). Together, these results show that loss of *ntc* causes previously undescribed phenotypes that

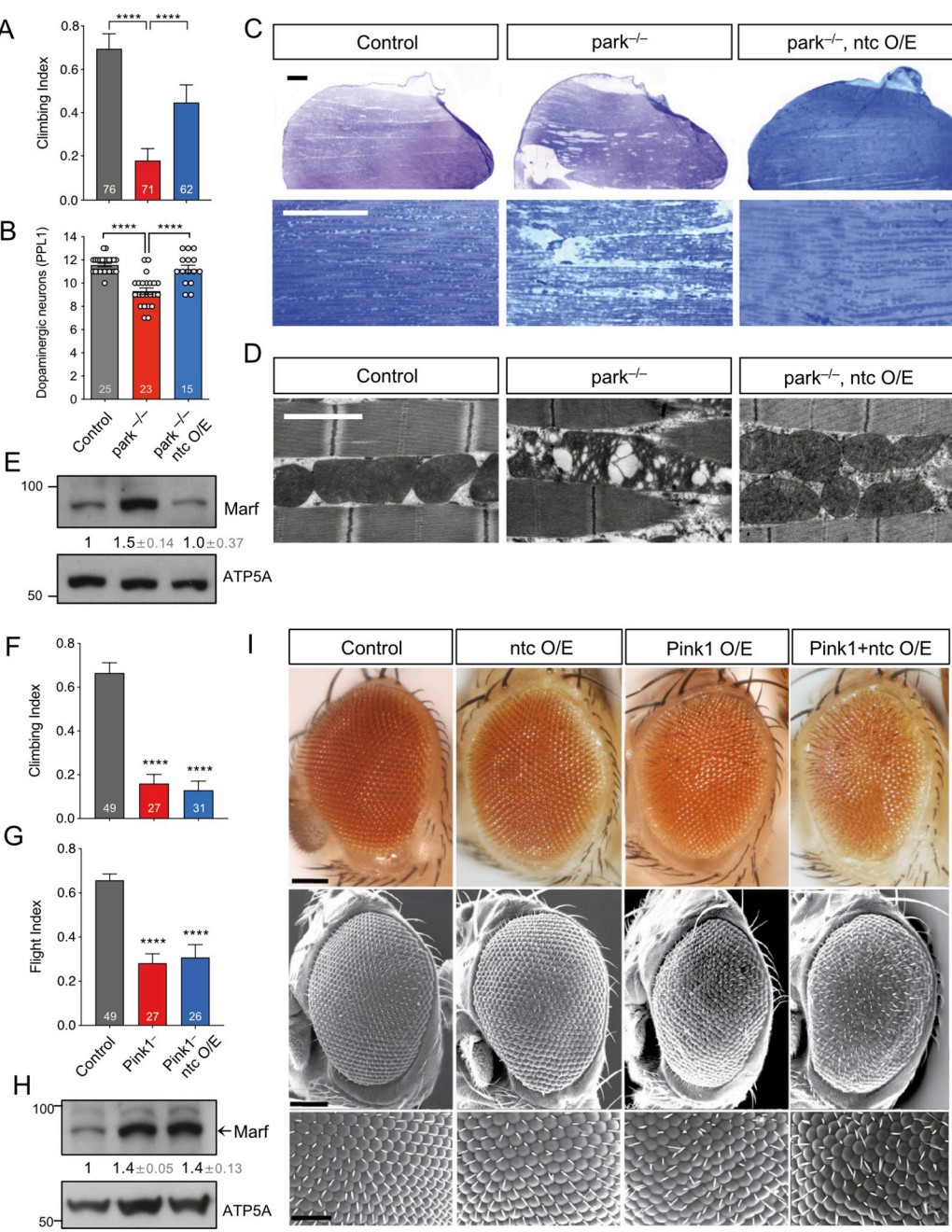

**Fig 1. ntc overexpression is able to rescue *parkin* but not *Pink1* mutants.** (**A**) Climbing ability of 2-day-old control and *parkin* mutants alone or with transgenic expression of *ntc* with the ubiquitous driver *da*-GAL4. Chart show mean ± 95% CI and *n* values. Kruskal–Wallis nonparametric test with Dunn's post hoc test correction for multiple comparisons; **** $P < 0.0001$. (**B**) Quantification of dopaminergic neurons (PPL1 cluster) in 30-day-old control and *parkin* mutants alone or with transgenic expression of *ntc* with the ubiquitous driver *da*-GAL4. Data represented as mean ± SEM; *n* shown in chart. One-way ANOVA with Bonferroni post hoc test correction; **** $P < 0.0001$. (**C**) Toluidine blue staining and (**D**) TEM analysis of thoraces from 2-day-old control and *parkin* mutant alone or with transgenic expression of *ntc* with the ubiquitous driver *da*-GAL4. Scale bars; top = 200 μm, middle = 10 μm, bottom = 2 μm. (**E**) Immunoblot analysis from whole fly lysates of Marf steady-state levels from 2-day-old control and *parkin* mutants alone or with transgenic expression of *ntc* with the ubiquitous driver *da*-GAL4. Numbers below blots indicated the mean ± SD of Marf levels normalised to ATP5A loading control across 3 replicate experiments. (**F**) Climbing and (**G**) flight ability of 2-day-old control and *Pink1* mutants alone or with transgenic expression of *ntc* with the ubiquitous driver *da*-GAL4. Chart show mean ± 95% CI; *n* shown in chart. Kruskal–Wallis nonparametric test with Dunn's post hoc test correction for multiple comparisons; **** $P < 0.0001$. (**H**) Immunoblot analysis from whole fly lysates of Marf steady-state levels from 2-day-old control and *Pink1*

mutants alone or with transgenic expression of *ntc* with the ubiquitous driver *da*-GAL4. Numbers below blots indicated the mean ± SD of Marf levels normalised to ATP5A loading control across 3 replicate experiments. (**I**) Analysis of the compound eye by (top panels) light microscopy or (middle and bottom panels) SEM of 2-day-old control, overexpression of ntc, Pink1 or both combine using the *GMR*-GAL4 driver. Scale bars; top and middle = 100 μm, bottom = 50 μm. Full details of numerical data and analyses underlying the quantitative data can be found in S1 Data. SEM, scanning electron microscopy; TEM, transmission electron microscopy.

impact organismal vitality similar to, but also different from, *Pink1/parkin* mutants, though the effects are generally less severe.

## ntc regulates basal mitophagy and antagonises USP30 in a Pink1/parkin-independent manner

Previous work showed that FBXO7 facilitates PINK1/Parkin-mediated mitophagy upon mitochondrial depolarisation in cultured cells [27]. Thus, we next sought to determine whether *Drosophila* ntc acts to promote mitophagy in vivo. The *mito*-QC reporter provides a sensitive and robust reporter for mitophagy whereby the GFP of OMM-localised tandem mCherry-GFP is quenched in the acidic lysosome, revealing mitolysosomes as "mCherry-only" puncta [35]. Importantly, in the absence of potent mitochondrial toxins—not readily applicable in vivo—the steady-state analysis of *mito*-QC provides an insight into basal mitophagy [20,36]. Quantifying basal mitophagy in larval neurons revealed a significant reduction in *ntc* mutants compared to controls, which was rescued upon re-expression of *ntc* (Fig 3A and 3B). Conversely, *ntc* overexpression significantly increased basal mitophagy (Fig 3C and 3D). To verify these results, we analysed an alternative mitophagy reporter with a mitochondrial matrix-targeted version of *mito-QC*, termed *mtx-QC*, with equivalent results (S1A and S1B Fig). Thus, ntc is both necessary and sufficient to promote basal mitophagy. Similarly, transgenic expression of *FBXO7* also increased mitophagy in larval neurons, underscoring their shared functionality (S2A and S2B Fig). Further validating these results in human cells stably expressing the *mito*-QC reporter [37], small interfering RNA (siRNA)-mediated knockdown of *FBXO7* also significantly reduced basal mitophagy (S2C–S2E Fig). Together, these results support a conserved function of FBXO7/ntc in regulating basal mitophagy.

While stress-induced mitophagy has been extensively studied, especially in cultured cell lines, mechanisms regulating basal mitophagy are poorly characterised in vivo. Under basal conditions loss of *USP30*, an established regulator of mitophagy, leads to accumulation of ubiquitinated OMM proteins, and sustained *USP30* loss or inhibition promotes mitophagy in vitro [38–42], but in vivo characterisation remains limited. Upon *USP30* knockdown, we observed a significant increase in basal mitophagy in larval neurons, using both the *mito-QC* (Fig 3E and 3F) and *mtx-QC* reporters (S1A and S1B Fig), and in adult flight muscle (S3A and S3B Fig). These results indicate that USP30 indeed inhibits basal mitophagy in vivo as expected. Importantly, we found that the induction of basal mitophagy by loss of *USP30* requires the activity of ntc as it was abolished in an *ntc* mutant background (Fig 3E and 3F).

As an orthogonal approach to test for genetic interaction at the organismal level, we found that while *USP30* knockdown did not cause any gross defect in adult viability or behaviour (S3C Fig), *USP30* overexpression caused a locomotor defect (Fig 3G). This phenotype was completely suppressed by co-expression of *ntc* (Fig 3G), consistent with the opposing actions of Ub-ligase and deubiquitinase and supporting the idea of ntc acting upstream of USP30. Interestingly, while *parkin* overexpression also suppressed the *USP30* overexpression phenotype as expected (S3D Fig), other Ub-ligases linked to mitophagy, Mul1 and March5 [39,43–47], showed no suppression of this phenotype (S3D Fig). Moreover, similar to *parkin* loss [20]

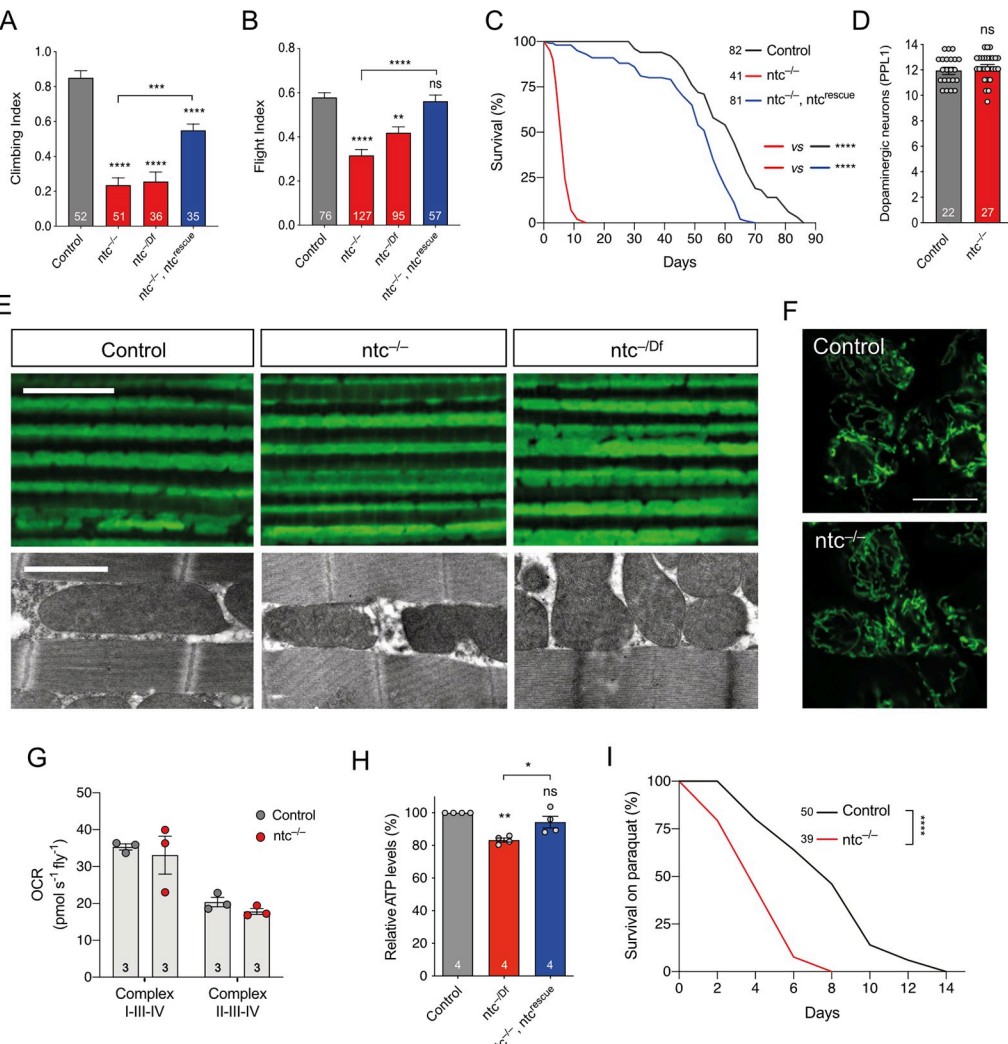

**Fig 2. Loss of ntc has motor and lifespan deficits, no gross effect on mitochondria but increased sensitivity to oxidative stress.** Analysis of (**A**) climbing and (**B**) flight abilities of 2-day-old control and *ntc* mutants alone or with transgenic expression of *ntc* with the ubiquitous driver *da*-GAL4. Chart show mean ± 95% CI and *n* values. Kruskal–Wallis nonparametric test with Dunn's post hoc test correction for multiple comparisons; ** $P < 0.01$, *** $P < 0.001$, **** $P < 0.0001$. (**C**) Lifespan assay of control and *ntc* mutants alone or with transgenic expression of *ntc* with the ubiquitous driver *da*-GAL4; *n* shown in chart. Log rank (Mantel–Cox) test; **** $P < 0.0001$. (**D**) Quantification of dopaminergic neurons (PPL1 cluster) in 10-day-old control and *ntc* mutants alone. Data represented as mean ± SEM; *n* shown in chart. One-way ANOVA with Bonferroni post hoc test correction. (**E**) (Top panel) Immunohistochemistry with anti-ATP5A staining and (bottom panel) TEM of 2-day-old adult thoraces from control and *ntc* mutants. Scale bars; top = 10 μm, bottom = 2 μm. (**F**) Mitochondrial morphology in motoneurons from the ventral nerve cord of third instar larvae in control and *ntc* mutants, overexpressing mito-GFP with the pan-neuronal driver *nSyb*-GAL4. Scale bars = 10 μm. (**G**) Oxygen consumption rate from complex I and complex II of 2-day-old control and *ntc* mutants. Data represented as mean ± SEM; *n* shown in chart. One-way ANOVA with Bonferroni post hoc test correction. (**H**) ATP levels of 2-day-old control and *ntc* mutants alone or with transgenic expression of *ntc* with the ubiquitous driver *da*-GAL4. Data represented as mean ± SEM; *n* shown in chart. One-way ANOVA with Bonferroni post hoc test correction; * $P < 0.05$, ** $P < 0.01$. (**I**) Lifespan assay of control and *ntc* mutants treated with 10 μm paraquat; *n* shown in chart. Log rank (Mantel–Cox) test; **** $P < 0.0001$. Full details of numerical data and analyses underlying the quantitative data can be found in S1 Data. TEM, transmission electron microscopy.

or overexpression (S3E and S3F Fig), loss of Mul1 or March5 had no effect on basal mitophagy (S3G–S3J Fig). Together, these data highlight the specificity of ntc and USP30 as important and opposing regulators of basal mitophagy.

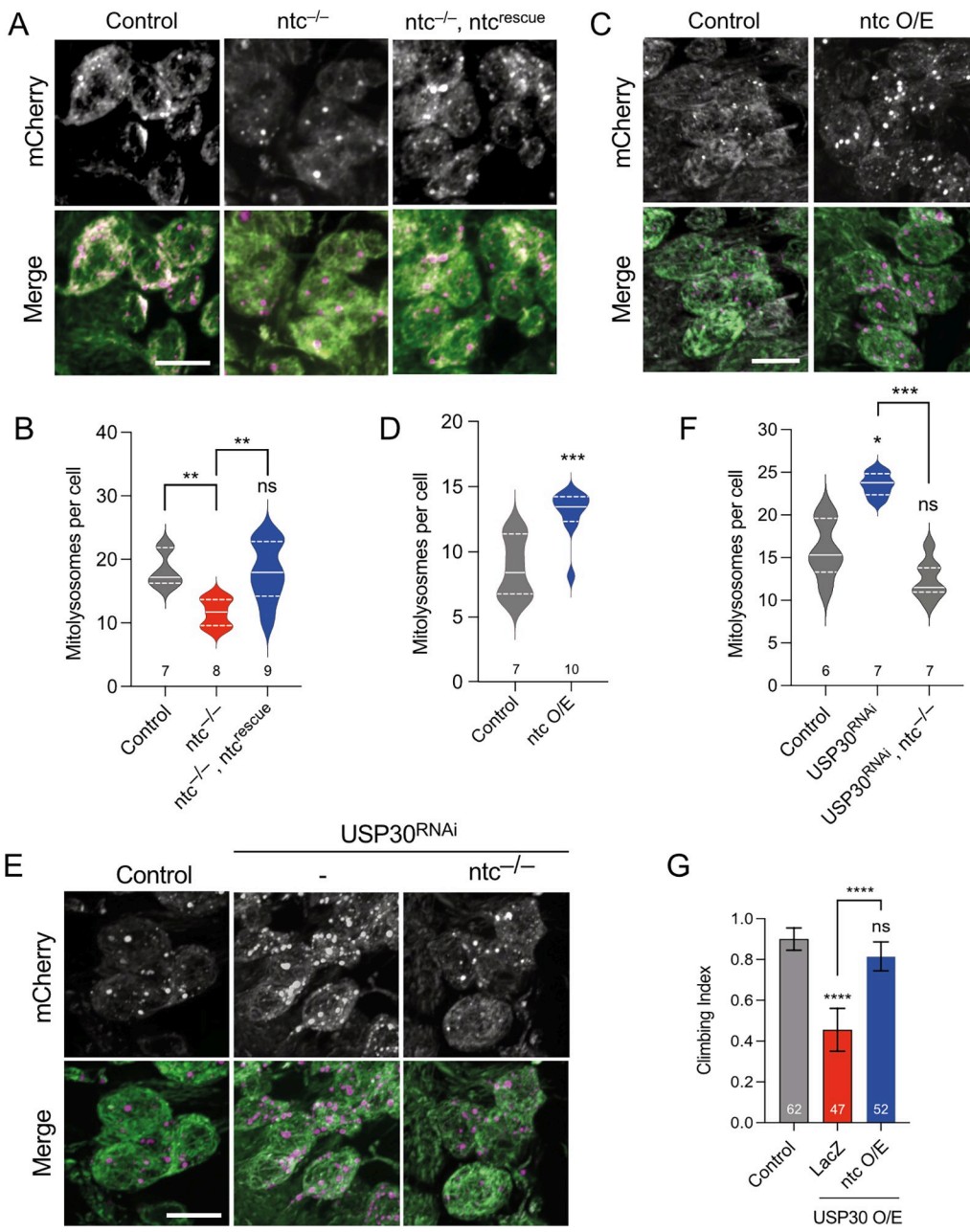

**Fig 3. ntc affects basal mitophagy and counteracts USP30.** (**A**, **B**) Confocal microscopy analysis of the *mito-QC* reporter in larval CNS of control and *ntc* mutant alone or with transgenic expression of *ntc* with the pan-neuronal driver *nSyb*-GAL4. Mitolysosomes are evident as GFP-negative/mCherry-positive (red-only) puncta; *n* shown in chart. One-way ANOVA with Bonferroni post hoc test correction; ** $P < 0.01$. Scale bars = 10 μm. (**C**, **D**) Confocal microscopy analysis of the *mito-QC* reporter in larval CNS of control and transgenic expression of *ntc* with the pan-neuronal driver *nSyb*-GAL4. Mitolysosomes are evident as GFP-negative/mCherry-positive (red-only) puncta; *n* shown in chart. One-way ANOVA with Bonferroni post hoc test correction; *** $P < 0.001$. Scale bars = 10 μm. (**E**, **F**) Confocal microscopy analysis of the *mito-QC* reporter in larval CNS of control and knockdown of USP30 with the pan-neuronal driver *nSyb*-GAL4 alone or in combination with *ntc* mutant. Mitolysosomes are evident as GFP-negative/mCherry-positive (red-only) puncta; *n* shown in chart. One-way ANOVA with Bonferroni post hoc test correction; * $P < 0.05$, *** $P < 0.001$. Scale bars = 10 μm. (**G**) Climbing ability of 10-day-old flies overexpressing USP30 alone or in combination with ntc or parkin with the ubiquitous driver *Act*-GAL4. Chart show mean ± 95% CI; *n* shown in chart. Kruskal–Wallis nonparametric test with Dunn's post hoc test correction for multiple comparisons; **** $P < 0.0001$. Full details of numerical data and analyses underlying the quantitative data can be found in S1 Data. CNS, central nervous system.

Given the previously established links between FBXO7 and USP30 with toxin-induced PINK1/Parkin mitophagy in cultured human cells, we next analysed whether the induction of basal mitophagy by *ntc* overexpression or *USP30* knockdown involved the *Pink1/parkin* pathway in vivo. As previously reported [20], the absence of *parkin* or *Pink1* did not impact basal mitophagy in larval neurons (Fig 4A–4D). However, overexpression of *ntc* was still able to induce mitophagy in the absence of either *parkin* or *Pink1* (Fig 4A–4D). Likewise, *USP30* knockdown also increased mitophagy independently of *parkin* and *Pink1* (Fig 5A–5D),

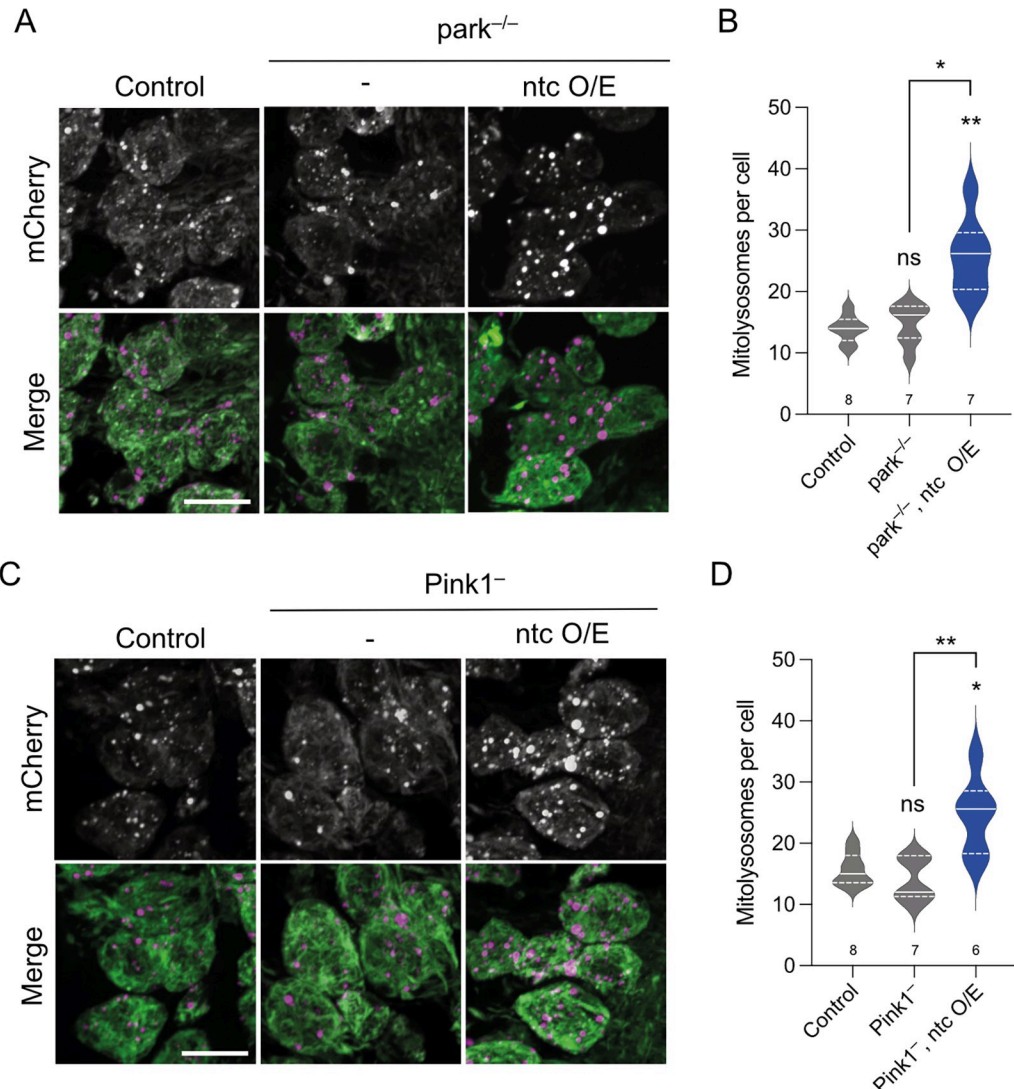

**Fig 4. ntc regulates basal mitophagy in a parkin- and Pink1-independent manner.** (**A**, **B**) Confocal microscopy analysis of the *mito-QC* reporter in larval CNS of control and *parkin* mutant alone or with transgenic expression of *ntc* with the pan-neuronal driver *nSyb*-GAL4. Mitolysosomes are evident as GFP-negative/mCherry-positive (red-only) puncta; *n* shown in chart. One-way ANOVA with Bonferroni post hoc test correction; * $P < 0.05$, ** $P < 0.01$. Scale bars = 10 μm. (**C**, **D**) Confocal microscopy analysis of the *mito-QC* reporter in larval CNS of control and *Pink1* mutant alone or with transgenic expression of *ntc* with the pan-neuronal driver *nSyb*-GAL4. Mitolysosomes are evident as GFP-negative/mCherry-positive (red-only) puncta; *n* shown in chart. One-way ANOVA with Bonferroni post hoc test correction; * $P < 0.05$, ** $P < 0.01$. Scale bar = 10 μm. Full details of numerical data and analyses underlying the quantitative data can be found in S1 Data. CNS, central nervous system.

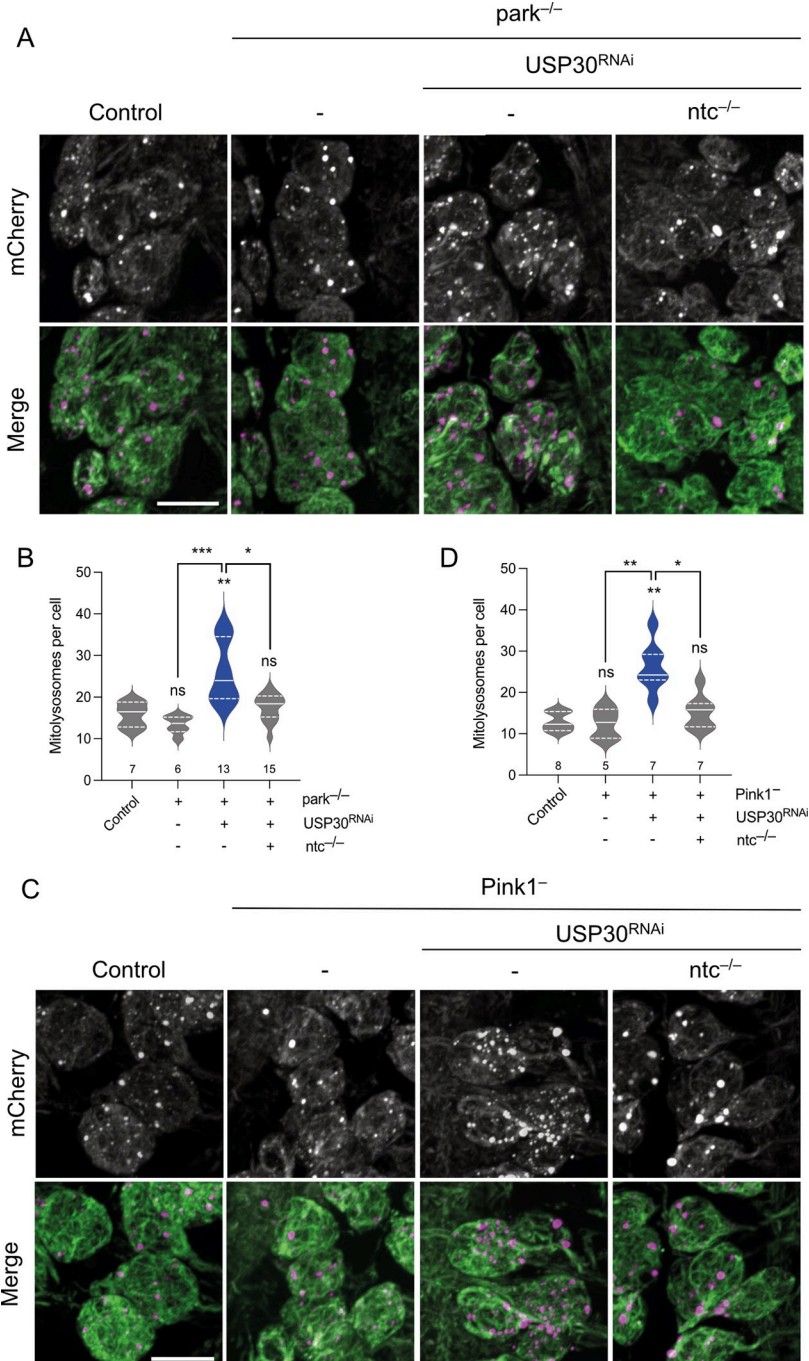

**Fig 5. ntc is required for USP30 knockdown induced basal mitophagy in the absence of parkin or Pink1. (A, B)**
Confocal microscopy analysis of the mito-QC reporter in larval CNS of control, knockdown of USP30 with the pan-neuronal driver *nSyb*-GAL4 alone or in combination with *ntc* mutants in a *parkin* mutant background. Mitolysosomes are evident as GFP-negative/mCherry-positive (red-only) puncta; *n* shown in chart. One-way ANOVA with Bonferroni post hoc test correction; * $P < 0.05$, ** $P < 0.01$, *** $P < 0.001$. Scale bars = 10 μm. (**C, D**) Confocal microscopy analysis of the *mito-QC* reporter in larval CNS of control, knockdown of USP30 with the pan-neuronal driver *nSyb*-GAL4 alone, or in combination with *ntc* mutants in a *Pink1* mutant background. Mitolysosomes are evident as GFP-negative/mCherry-positive (red-only) puncta; *n* shown in chart. One-way ANOVA with Bonferroni post hoc test correction; * $P < 0.05$, ** $P < 0.01$. Scale bar = 10 μm. Full details of numerical data and analyses underlying the quantitative data can be found in S1 Data. CNS, central nervous system.

consistent with in vitro findings [38,39,48,49]. Notably, the induction of mitophagy by *USP30* knockdown in *Pink1* or *parkin* mutant backgrounds required the function of *ntc* (Fig 5A–5D), further underscoring the importance of ntc in basal mitophagy.

## General autophagy is not grossly affected upon USP30 or ntc manipulations

Although previously validated as a reliable mitophagy reporter in vivo, we nevertheless verified that the increased mitolysosome formation (mitophagy) by *ntc* overexpression or *USP30* knockdown occurs via canonical autophagy flux as it was abrogated in the absence of *Atg8a*, the main fly homologue of human ATG8 family members GABARAP/LC3, in both wild-type or *parkin* mutant backgrounds (S4A–S4D Fig). Notwithstanding, the observed increase in mitolysosomes could be due to an increase in nonselective autophagy [50,51]. Thus, we used a number of well-established orthogonal assays to assess the effect of these mitophagy-inducing manipulations on general autophagy.

First, we examined the level of lipidated Atg8a (Atg8a-II) that is incorporated into autophagosomal membranes and is used as an indication of autophagy induction [52,53]. Neither *USP30* knockdown, *ntc* overexpression nor *ntc* loss had any effect on autophagy induction (Fig 6A and 6B). Similarly, the steady-state levels of ref(2)P, the homologue of mammalian p62, which accumulates upon autophagic blockage [54], was also not affected in any of the conditions (Fig 6C and 6D). We also analysed the autophagy flux reporter GFP-mCherry-Atg8a [55–57]. Quantification of mCherry-Atg8a puncta (autolysosomes) showed no changes upon manipulating levels of *USP30* or *ntc* (Fig 6E and 6F). Together, these results support that the effect observed upon manipulation of *USP30* or *ntc* is specific for basal mitophagy and not a consequence of altered general autophagic flux.

## ntc increases basal mitochondrial ubiquitin and promotes phospho-ubiquitin formation

To gain a mechanistic insight into the regulation of basal mitophagy by ntc and USP30, we analysed their impact on mitochondrial ubiquitination. Consistent with their molecular functions, both *ntc* overexpression and *USP30* knockdown increased the total amount of ubiquitin present in mitochondria-enriched fractions under basal conditions (Figs 7A and 7B and S5A), while total Ub remained unchanged (S5B Fig). Indeed, recent studies indicate that USP30 acts to remove preexisting OMM ubiquitin under basal conditions, influencing the threshold needed for Parkin activation [38,58].

We have recently described that loss of *parkin* causes a significant accumulation of pS65-Ub, consistent with parkin promoting the degradation of pS65-Ub labelled mitochondria [59]. Unlike loss of *parkin*, loss of *ntc* alone did not lead to pS65-Ub build up (Fig 7C and 7D). However, *parkin*:*ntc* double mutants result in a significant reduction in the accumulated pS65-Ub levels (Fig 7C and 7D), and overexpression of *ntc* in the *parkin* mutant background increased the amount of pS65-Ub compared to *parkin* mutants alone (Fig 7E and 7F). These results are consistent with ntc promoting basal mitochondrial ubiquitination, which is phosphorylated by Pink1 to drive mitophagy. Thus, together these data suggest that ntc and USP30 work antagonistically to set up an OMM ubiquitin threshold needed for subsequent mitochondrial clearance.

## Discussion

Mutations in *FBXO7* have been linked to familial parkinsonism but the mechanisms of pathogenesis are poorly understood [22,23]. We have characterised the putative *Drosophila*

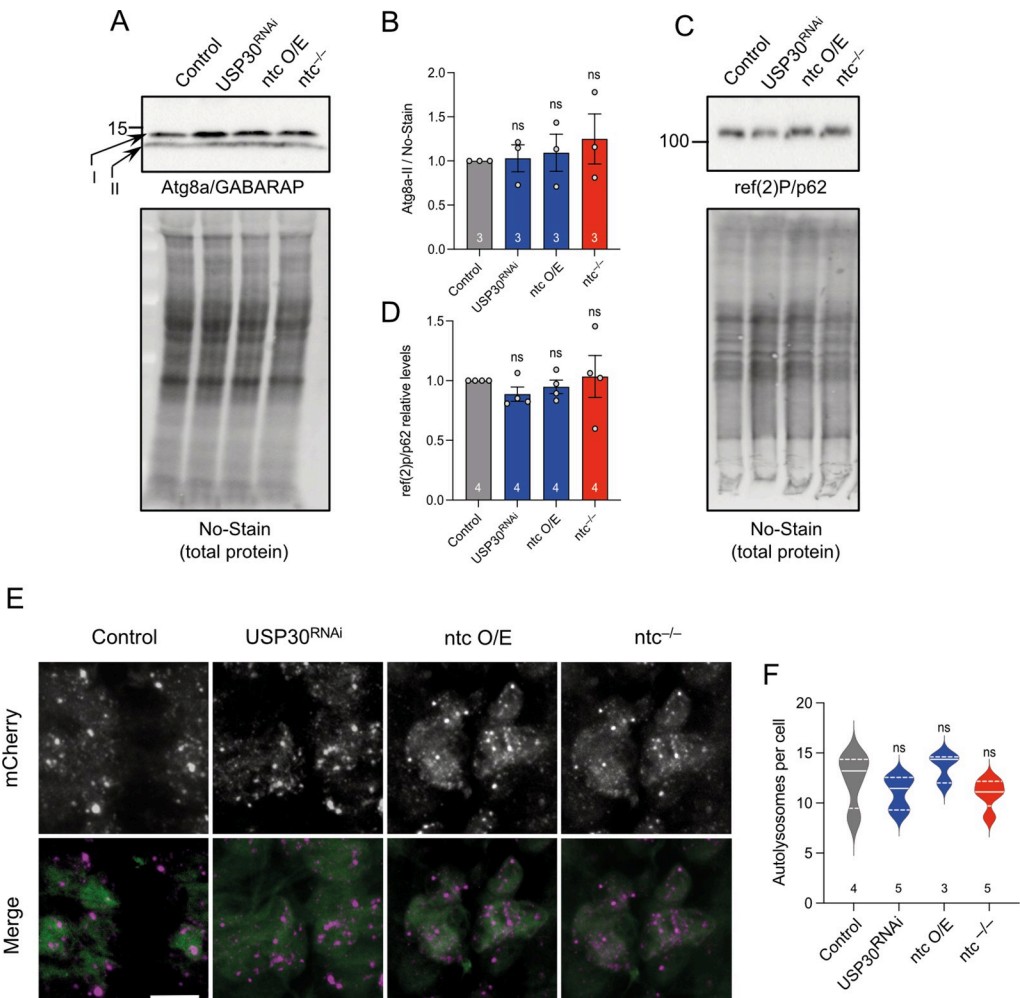

**Fig 6. Manipulating mitophagy by altering levels of USP30 or ntc do not have an effect on general autophagy.** (**A**, **B**) Representative immunoblot from whole fly lysates of Atg8a (non-lipidated) I and II (lipidated) in control, USP30 knockdown, ntc overexpression, and *ntc* mutant with the ubiquitous driver *da*-GAL4. Data represented as mean ± SD; *n* shown in chart. One-way ANOVA with Bonferroni post hoc test correction. (**C**, **D**) Representative immunoblot from whole fly lysates of ref(2)P/p62 in control, USP30 knockdown, ntc overexpression, and *ntc* mutant with the ubiquitous driver *da*-GAL4. Data represented as mean ± SD; *n* shown in chart. One-way ANOVA with Bonferroni post hoc test correction. (**E**, **F**) Confocal microscopy analysis and quantification of the red-only (autolysosomes) puncta per cell of larval CNS expressing the autophagy flux marker GFP-mCherry-Atg8a in combination with USP30 knockdown, ntc overexpression, and *ntc* mutant with the pan-neuronal driver *nSyb*-GAL4. Data represented as mean ± SEM; *n* shown in chart. One-way ANOVA with Bonferroni post hoc test correction; scale bar = 10 μm. Full details of numerical data and analyses underlying the quantitative data can be found in S1 Data. CNS, central nervous system.

homologue, *ntc*, to assess the functional homology with *FBXO7* and its potential as a model to understand the function of FBXO7 in vivo.

FBXO7 has been genetically linked to *Drosophila Pink1* and *parkin* and shown to facilitate PINK1/Parkin-mediated mitophagy in vitro. The *Drosophila* models have provided valuable insights into Pink1/parkin biology and genetic interactors have highlighted complementary pathways in mitochondrial homeostasis. Here, we show that *ntc* mutants partially phenocopy *Pink1/parkin* phenotypes in locomotor function, lifespan, and sensitivity to paraquat, consistent with a role in mitochondria quality control. We also observed the same genetic relationship as previously reported for *FBXO7* [27], namely, that overexpression of *ntc* can rescue all

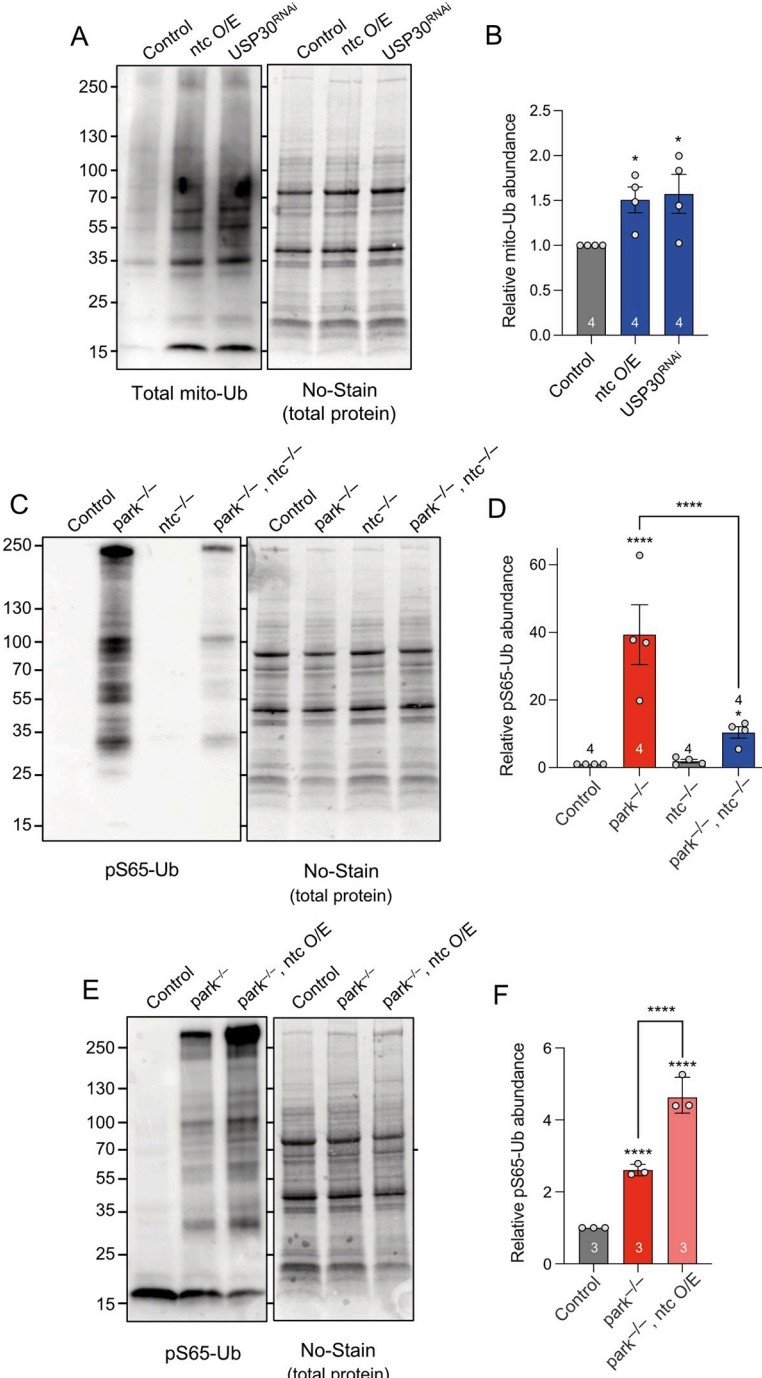

**Fig 7. Both ntc overexpression and USP30 depletion promote accumulation of ubiquitin in the mitochondria and are necessary for pS65-Ub formation.** (**A**) Representative immunoblot and quantification of (**B**) mitochondrial ubiquitin (P4D1) in control, ntc overexpression and USP30 knockdown with the ubiquitous driver *da*-GAL4. Data represented as mean ± SD; *n* shown in chart. One-way ANOVA with Bonferroni post hoc test correction; * $P < 0.05$. (**C, D**) Representative immunoblot and quantification of mitochondrial pS65-Ub in *parkin* mutants, *ntc* mutants, and the double *parkin*:*ntc* mutant. Data represented as mean ± SD; *n* shown in chart. One-way ANOVA with Bonferroni post hoc test correction; * $P < 0.05$, **** $P < 0.0001$. (**E, F**) Representative immunoblot and quantification of mitochondrial pS65-Ub in control and *parkin* mutants alone or with the transgenic expression of ntc with the ubiquitous driver *da*-GAL4. Data represented as mean ± SD; *n* shown in chart. One-way ANOVA with Bonferroni post hoc test correction; **** $P < 0.0001$. Full details of numerical data and analyses underlying the quantitative data can be found in S1 Data.

*parkin* mutant phenotypes but not for *Pink1* mutants. It is worth noting that in our previous study [27], we performed an initial characterisation of *ntc* as the putative homologue of *FBXO7* but observed no phenotypes reminiscent of *Pink1/parkin* mutants and provisionally concluded that ntc was not a functional homologue of FBXO7. However, these analyses were conducted using a hypomorphic allele (*ntc^f07259^*) in homozygosity, weakening the phenotype penetrance. In the current study, we have performed our analyses with an amorphic allele, in parallel to studies comparing transgenic expression of *ntc* with *FBXO7*, hence, supporting a functional homology between ntc and FBXO7.

Importantly, FBXO7 has previously been shown to facilitate the PINK1/Parkin pathway to promote stress-induced mitophagy in vitro [27,60]. However, whether it regulates mitophagy in vivo is poorly characterised. Indeed, mitophagy itself can occur via different regulators and stimuli and represents only one of several mechanisms of mitochondrial quality control [61,62]. For example, most studies have utilised a mitochondrial toxification stimulus to model substantial "damage;" however, basal mitophagy occurs as a housekeeping-type mechanism to regulate mitochondrial quantity or piecemeal turnover. The development of genetically encoded mitophagy reporters has provided a powerful approach to monitor both stress-induced and basal mitophagy in vivo. We have shown here that ntc is able to modulate basal mitophagy in a Pink1/parkin-independent manner, which is conserved by FBXO7 in human cell lines. Interestingly, a recent study emphatically described FBXO7 as performing no role in stress-induced Parkin-dependent mitophagy [63], though basal mitophagy was not specifically analysed. Relevant to this, additional work from our group [64] and others [65] has shown that basal mitophagy increases with age, presumably in response to age-related stresses, which is selectively affected by *Pink1/parkin* highlighting a physiologically relevant stress. The fact that FBXO7 does not robustly affect acute toxin-induced mitophagy in cultured cells [63] likely reflects the exaggerated and non-physiological nature of this approach that masks more subtle, physiologically relevant processes.

In comparison to stress-induced mitophagy, the regulation of basal mitophagy is poorly characterised, particularly in metazoa. While early studies showed the mitochondrial deubiquitinase USP30 antagonises PINK1/Parkin-mediated mitophagy [41,66,67], recent studies have established a role in basal mitophagy [38–40,42]. The emerging view suggests the main role of USP30 is acting as part of a quality control process for intra-mitochondrial proteins during import through the translocon [39,42] and that basal mitochondrial ubiquitination provides the initiating substrate for PINK1 signalling. This paradigm requires an Ub ligase to provide basal ubiquitination that USP30 acts upon. While Phu and colleagues [42] suggested that this may be mediated via March5, this was not concurred by Ordureau and colleagues [39]. Thus, the ligase providing the basal ubiquitination is unclear, and the role of USP30 in basal mitophagy in vivo remained to be established.

Here, we demonstrated that knockdown of USP30 in vivo does indeed increase basal mitophagy in neurons and muscle, consistent with the in vitro studies. This also established a paradigm for a more deeply analysis of ntc's function in basal mitophagy. While the induction of basal mitophagy by *USP30* knockdown is Pink1/parkin-independent as expected, it nevertheless required the function of ntc. In contrast, overexpression of parkin or depletion of either Mul1 or March5 did not affect neuronal basal mitophagy in our assay. The antagonistic relationship between USP30 and parkin was also evident from genetic interactions at the organismal level (rescue of a climbing defect), as expected, and was recapitulated with ntc. Such an interaction was also not evident with Mul1 and March5. Mechanistically, we found that *ntc* promotes mitochondrial ubiquitination under basal conditions, which was mirrored by the down-regulation of *USP30*, consistent with the idea that basal ubiquitination signals basal

mitophagy. Thus, our data suggest that ntc/FBXO7 provides basal ubiquitination that is antagonised by USP30, which facilitates basal mitophagy upon a yet unclear stimulus.

The current mechanistic view of PINK1/Parkin stress-induced mitophagy signalling posits that upon mitochondrial depolarisation or import blocking, PINK1 becomes stabilised in the translocon, phosphorylates latent, preexisting Ub, which promotes the recruitment of Parkin, triggering the feed-forward mechanism. Ubiquitination of import substrates would provide a ready source of latent Ub for PINK1 to phosphorylate as/when it becomes activated. Interestingly, USP30 has been found to be associated with TOMM20, a bona fide substrate of FBXO7 [68], and other translocon assemblies, where basally ubiquitinated translocon import substrates accumulate [39]. Consistent with ntc providing the basal ubiquitination in this model, pS65-Ub that accumulates in *parkin* mutants is both reduced by loss of *ntc* and increased by its overexpression.

While we have shown that *ntc* overexpression is sufficient to induce basal mitophagy, independently of *Pink1/parkin*, and it is able to rescue *parkin* mutant phenotypes, curiously, it is not able to rescue *Pink1* phenotypes. As forced overexpression of *ntc* increases basal ubiquitination, it follows that already high levels of substrate for Pink1 circumvents the need for parkin ubiquitination activity (hence, why *ntc* overexpression can substitute for loss of *parkin*) when stress-induced mitophagy is required. However, despite high levels of latent ubiquitination, this alone is not sufficient to trigger stress-induced mitophagy in the absence of Pink1-mediated phosphorylation (hence, why *ntc* overexpression cannot rescue *Pink1* mutants). These findings further underscore the mechanistic and functional differences between basal and stress-induced mitophagy.

Analysing the potential role of ntc/FBXO7 in mitophagy and the relationship with Pink1/parkin in vivo has highlighted the complexity of how different forms of mitophagy may influence tissue homeostasis and relate to organismal phenotypes. As stated before, there are many different forms of mitochondrial quality control and mitophagy, and these perform different functions in cellular remodelling and homeostasis [61,62]. It follows that some forms of mitophagy may dramatically impact neuromuscular homeostasis when disrupted, while others may not. For example, evidence supports that PINK1/Parkin promote stress-induced mitophagy but are minimally involved in basal mitophagy, and *Drosophila* mutants present multiple robust phenotypes. In contrast, ntc/FBXO7 (and USP30) regulate basal mitophagy and facilitate PINK1/Parkin mitophagy by providing the initiating ubiquitination, yet their mutant phenotypes only loosely resemble those of *Pink1/parkin* mutants.

How do these forms of mitophagy correlate with the mitochondrial/organismal phenotypes? The *Pink1/parkin* phenotypes are consistent with a catastrophic loss of integrity in energy-intensive, mitochondria-rich tissues caused by the lack of a crucial protective measure (induced mitophagy) for a specific circumstance (likely, mitochondrial "damage" arising from a huge metabolic burst). In contrast, while loss of *ntc* causes a partial (but not complete) loss of basal mitophagy, the tissues affected in *Pink1/parkin* mutants appear to be able to cope with loss of this "housekeeping" mitochondrial QC process, consistent with there being partially redundant pathways. Notably, these tissues are still able to mount a stress-induced response via Pink1/parkin as we have observed paraquat-induced accumulation of pS65-Ub in the *ntc* mutant (S5C Fig), reinforcing that stress-induced mitophagy can happen in the absence of ntc. Consequently, *ntc* mutants do not present the same degenerative phenotypes as *Pink1/parkin* mutants and maintain relatively normal mitochondrial function. Hence, it seems likely that the additional *ntc* mutant phenotypes of motor deficits and drastically short lifespan are due to additional functions of ntc/FBXO7, such as regulation of proteasome function and caspase activation, and not mitophagy.

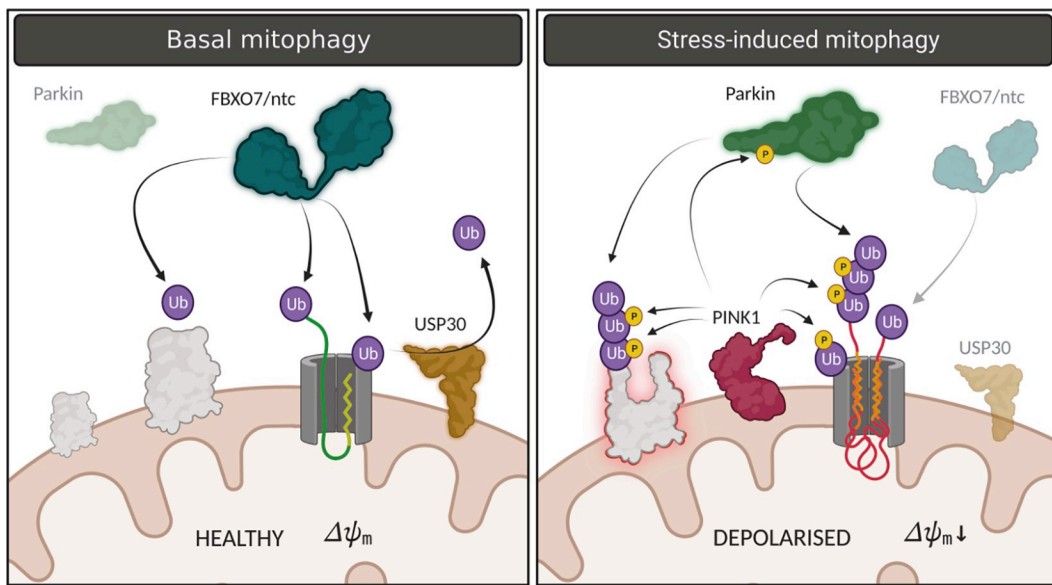

**Fig 8. Proposed model for basal and stress-induced mitophagy regulated by ntc, USP30, and parkin.** In basal, healthy conditions, FBXO7/ntc acts to prime OMM proteins with ubiquitin counteracted by USP30. This provides a key surveillance checkpoint in quality control of mitochondrial protein import and/or protein damage. Occasional defects in import or protein misfolding may lead to an Ub threshold that provokes basal mitophagy. However, when the need arises in the presence of a stress, for example, where the membrane may become depolarised, this population of latent Ub could subsequently be phosphorylated by Pink1 and amplified by parkin to trigger selective degradation upon specific stimulation. Created with BioRender.com. OMM, outer mitochondrial membrane.

In summary, we propose a model (Fig 8) in which FBXO7/ntc acts to prime OMM proteins with ubiquitin that is counteracted by USP30. This provides a key regulatory checkpoint in quality control, for example, during mitochondrial protein import. It is likely that occasional defects in import or protein misfolding may lead to an increase of Ub above a threshold that is sufficient to provoke basal mitophagy. However, when the need arises this population of latent Ub could subsequently be phosphorylated by PINK1 (and amplified by Parkin) to trigger selective degradation upon specific stimulation. Although more studies are required to understand the interplay between FBXO7/ntc, USP30, Pink1, and parkin, this study provides a foundation to further elucidate the interplay between these mechanisms of mitochondrial quality control and reinforces its potential as a therapeutic target.

## Materials and methods

### *Drosophila* stocks and procedures

*Drosophila* were raised under standard conditions in a temperature-controlled incubator with a 12 h:12 h light:dark cycle at 25°C and 65% relative humidity, on food consisting of agar, cornmeal, molasses, propionic acid, and yeast. *ntc*$^{ms771}$, *UAS-ntc*, and *PBac{WH} CG10855*$^{f07259}$ strains were kindly provided by H. Steller. *park*$^{25}$ mutants and *UAS-parkin*$_{C2}$ [13], *UAS-FBXO7* [27], and *UAS-mito-QC* lines [20] have been previously described. *Pink1*$^{B9}$ mutants [12] were provided by J. Chung (SNU). *UAS-Mul1* and *Mul1*$^{A6}$ mutants were kindly provided by Ming Guo. *UAS-USP30* lines were kindly provided by Ugo Mayor. *UAS-USP30* RNAi (NIG-Fly 3016R[II]) was obtained from the NIG-Fly collection. The following strains were obtained from Bloomington *Drosophila* Stock Center (BDSC, RRID:SCR_006457): *w*$^{1118}$ (RRID:BDSC_6326), *da-GAL4* (RRID:BDSC_55850), *nSyb-GAL4* (RRID:BDSC_51635),

*GMR*-GAL4 (RRID:BDSC_1104), *Mef2*-GAL4 (RRID:BDSC_27390), *Act*-GAL4 (RRID: BDSC_25374), *Atg8a* [KG07569] *(Atg8a$^{-/-}$)* (RRID:BDSC_14639), *UAS-GFP-mCherry-Atg8a* (RRID:BDSC_37749), and *Df(3L)Exel6097* (RRID:BDSC_7576). *UAS-lacZ* was obtained from FlyORF (RRID: FlyBase_FBst0503118). The following strains were obtained from Vienna Drosophila Research Centre (VDRC): *UAS-LacZ$^{RNAi}$* (v51446), *UAS-March5* RNAi KK (v105711), and *UAS-March5* RNAi GD(v33309). *UAS-mtx-QC* and *UAS-March5* lines were generated as follows: *mtx-QC* mCherry-GFP was amplified from *UAS-mito-QC* and cloned in-frame with the mitochondrial targeting sequence from hCOX8A into pUAST.attB and inserted in attP40 and attP16 sites; while for *March5* (*CG9855*), full-length cDNA was cloned into pUAST vector and transgenesis performed by random insertion. A full description of the genotypes used in this study is shown in S1 Table.

## Locomotor and lifespan assays

For locomotor assays, climbing (negative geotaxis assay) was assessed as previously described, with minor modifications [13]. For lifespan experiments, flies were grown under identical conditions at low density. Progeny were collected under very light anaesthesia and kept in tubes of approximately 20 males each. Flies were transferred every 2 to 3 days to fresh tubes with normal food for normal lifespan and tubes with 10 mM paraquat in a filter paper for the oxidative stress lifespan. The number of dead flies was recorded on each transfer. Percent of survival was calculated at the end of the experiment after correcting for any accidental loss.

## Histology of adult thoraces

Thoraces were prepared from 5-day-old adult flies and treated as previously described [13]. Semi-thin sections were then taken and stained with Toluidine blue, while ultra-thin sections were examined using a FEI Tecnai G2 Spirit 120KV transmission electron-microscope.

## Light microscopy imaging and scanning electron microscopy of *Drosophila* eye

Light microscopy imaging was assessed using a Nikon motorised SMZ stereo zoom microscope fitted with 1× Apo lens. Extended focus images were then generated using Nikon Elements software using the same settings for all the genotypes. Flies were anaesthetised with $CO_2$ during the process. Scanning electron microscopy (SEM) was performed according to a standard protocol [69]. All animals of a given genotype displayed essentially identical phenotypes and randomly selected representative images are shown. Images were taken using an SEM microscope (Philips XL-20 SEM).

## Immunohistochemistry and sample preparation

*Drosophila* brains were dissected from 30-day-old flies and immuno-stained with anti-tyrosine hydroxylase (Immunostar Inc. #22491) as described previously [14]. Brains were imaged with an Olympus FV1000 confocal with SIM-scanner on a BX61 upright microscope. Tyrosine hydroxylase-positive neurons were counted under blinded conditions. For immunostaining, adult flight muscles were dissected in PBS and fixed in 4% formaldehyde for 30 min, permeabilized in 0.3% Triton X-100 for 30 min, and blocked with 0.3% Triton X-100 plus 1% bovine serum albumin in PBS for 1 h at RT. Tissues were incubated with ATP5A antibody (Abcam Cat# ab14748, RRID:AB_301447; 1:500), diluted in 0.3% Triton X-100 plus 1% bovine serum albumin in PBS overnight at 4°C, then rinsed 3 times 10 min with 0.3% Triton X-100 in PBS, and incubated with the appropriate fluorescent secondary antibodies overnight at 4°C. The

tissues were washed 3 times in PBS and mounted on slides using ProLong Diamond Antifade mounting medium (Thermo Fisher Scientific) and image next day. For mitolysosome analysis of *mito-QC* and *mtx-QC*, tissues were dissected and treated as previously described [20].

## Mitochondrial morphology in larval brain

Third instar larvae overexpressing *UAS-mitoGFP* with the pan-neuronal driver nSyb-GAL4 were dissected in PBS and fixed in 4% formaldehyde for 30 min. The tissues were washed 3 times in PBS and mounted on slides using ProLong Diamond Antifade mounting medium (Thermo Fisher Scientific) and image next day in a Zeiss LSM880 confocal microscope (63×/ 1.4 NA).

## ATP levels

The ATP assay was performed as described previously [70]. Briefly, 5 male flies of the indicated age for each genotype were homogenised in 100 μL 6M guanidine-Tris/EDTA extraction buffer and subjected to rapid freezing in liquid nitrogen. Homogenates were diluted 1/100 with the extraction buffer and mixed with the luminescent solution (CellTiter-Glo Luminescent Cell Viability Assay (Promega, RRID:SCR_006724)). Luminescence was measured with a SpectraMax Gemini XPS luminometer (Molecular Devices). The average luminescent signal from technical triplicates was expressed relative to protein levels, quantified using the DC Protein Assay kit (Bio-Rad Laboratories, RRID:SCR_008426). Data from 2 to 4 independent experiments were averaged and the luminescence expressed as a percentage of the control.

## Respirometry

Mitochondrial respiration was assayed at 30°C by high-resolution respirometry using an Oxygraph-2k high-resolution respirometer (OROBOROS Instruments) using a chamber volume set to 2 mL. Calibration with the air-saturated medium was performed daily. Data acquisition and analysis were carried out using Datlab software (OROBOROS Instruments). Five flies per genotype were homogenised in Respiration Buffer [120 mM sucrose, 50 mM KCl, 20 mM Tris–HCl, 4 mM $KH_2PO_4$, 2 mM $MgCl_2$, and 1 mM EGTA, 1 g $L^{-1}$ fatty acid-free BSA (pH 7.2)]. For coupled (state 3) assays, complex I-linked respiration was measured at saturating concentrations of malate (2 mM), glutamate (10 mM), L-proline (10 mM), and adenosine diphosphate (ADP, 2.5 mM). Complex II-linked respiration was assayed in Respiration Buffer supplemented with 0.15 μm rotenone, 10 mM succinate, and 2.5 mM ADP. Respiration was expressed as oxygen consumed per fly. Flies' weight was equal in all genotypes tested.

## Image analysis and quantification of mitolysosomes

Analysis of mitolysosomes was done as previously described [20]. Briefly, spinning disk microscopy-generated images from dissected larval brains or adult thoraces were processed using Imaris (version 9.0.2) analysis software (BitPlane; RRID:SCR_007370) to identify and count individual red-only puncta. The GFP and mCherry signals were adjusted to reduce background noise and retain only the distinct mitochondria network and red puncta, respectively. A surface rendered 3D structure corresponding to the mitochondria network was generated using the GFP signal. This volume was subtracted from the red channel to retain the mCherry signal that did not colocalize with the GFP-labelled mitochondria network. The mitolysosomes puncta were selected according to their intensity and an estimated size of 0.5 μm diameter, previously measured with Imaris. Additionally, the puncta were filtered with a minimum size cut off of 0.2 μm diameter. The remaining puncta were counted as the

number of mitolysosomes. Larval CNS soma were analysed individually where discrete cells could be distinguished. The mean number of mitolysosomes per cell was calculated per animal. Data points in the quantification charts show mean mitolysosomes per cell for individual animals, where $n \geq 6$ animals for each condition.

### Analysis and quantification of autolysosomes using the GFP-mCherry-Atg8a traffic light reporter

Third instar larval brains were dissected in phosphate buffered saline (PBS) and fixed with 4% formaldehyde (FA) pH 7 (Thermo Scientific)/PBS for 20 min at room temperature. Samples were then washed in PBS followed by water to remove salts. ProLong antifade mounting media (Thermo Scientific) was used to mount the samples and imaged the day after. Confocal images, acquired with a Zeiss LSM880 microscope with the 63×/1.4 NA oil, were processed using FIJI (Image J). The quantification of autolysosomes was performed using FIJI (Image J) with the 3D Objects Counter Plugin. An area of interest was selected by choosing 6 to 10 cells per image. The threshold was based on matching the mask with the fluorescence. All puncta larger than 0.049 $\mu m^3$ was considered an autolysosome. Data points in the quantification charts show mean mitolysosomes per cell for individual animals, where $n \geq 3$ animals for each condition.

### Mitochondrial enrichment by differential centrifugation

All steps were performed on ice or at 4°C. For immunoblotting analysis and biochemical fractionation from small numbers of flies (10–30), a modified mitochondrial enrichment procedure was performed. Flies were prepared either fresh or after flash-freezing in liquid nitrogen, with all direct comparisons performed with flies that were prepared in the same manner. Flies were transferred into a Dounce homogeniser containing 700 μL Solution A (70 mM sucrose, 20 mM HEPES (pH 7.6), 220 mM mannitol, 1 mM EDTA) containing cOmplete protease inhibitors (Roche) and PhosSTOP phosphatase inhibitors (Roche), and manually homogenised with 50 strokes of a pestle. The homogenate was transferred to an Eppendorf tube, a further 500 μL of Solution A was added to the homogeniser and the sample was homogenised with a further 10 strokes. The homogenates were pooled and incubated for 30 min, then centrifuged for 5 min at 800 × g. The supernatant (containing mitochondria) was transferred to a new tube and clarified twice by centrifugation for 5 min at 1,000 × g. The clarified supernatant was then centrifuged for 10 min at 10,000 × g, and the post-mitochondrial supernatant was discarded and the pellet retained for analysis. The mitochondrial pellet was washed once in Solution A containing only protease inhibitors, and then once in Solution A without inhibitors. The washed mitochondrial pellet was resuspended in 50 to 200 μL Sucrose Storage Buffer, the protein content determined by BCA assay (Thermo Pierce) and stored at −80°C until needed.

### Antibodies and dyes

The following mouse antibodies were used for immunoblotting (WB) in this study: ATP5A (Abcam, ab14748, 1:10,000), actin (Millipore, MAB1501, 1:5,000), Ubiquitin (clone P4D1, Cell Signalling Technology, 3936, 1:1,000), Ubiquitin (clone FK2, MBL, D058-3, 1:1,000). The following rabbit antibodies were used in this study: pS65-Ub (Cell Signalling Technologies, 62802S, 1:1,000), Marf ([16], 1:1,000), GABARAP/Atg8a (Abcam ab109364, 1:1,000), ref(2)P/p62 (Abcam ab178440, 1:1,000). The following secondary antibodies were used: sheep anti-mouse (HRP-conjugated, GE Healthcare, NXA931V, 1:10,000), donkey anti-rabbit (HRP-conjugated, GE Healthcare, NA934V, 1:10,000).

## Whole-animal lysis and immunoblotting

For the analysis of whole cell lysates by immunoblot, 180 μL cold RIPA buffer (150 mM NaCl, 1% (v/v) NP-40, 0.5% (w/v) sodium deoxycholate, 0.1% (w/v) SDS, 50 mM Tris (pH 7.4)), supplemented with cOmplete protease inhibitors, was added to 2 mL tubes containing 1.4 mm ceramic beads (Fisherbrand 15555799). Animals (5 to 20 per replicate) were harvested and stored on ice or flash-frozen in liquid $N_2$, with all direct comparisons performed with flies that were harvested in the same manner. The flies were added to the tubes containing RIPA buffer and lysed using a Minilys homogeniser (Bertin Instruments) with the following settings: maximum speed, 10 s on, 10 s on ice, for a total of 3 cycles. After lysis, samples were returned to ice for 10 min then centrifuged 5 min at 21,000 × g, 4˚C. A total of 90 μL supernatant was transferred to a fresh Eppendorf tube and centrifuged a further 10 min at 21,000 × g. Approximately 50 μL supernatant was then transferred to a fresh Eppendorf tube and the protein content determined by BCA assay as above, and 30 μg total protein was then diluted in Laemmli Sample Buffer (Bio-Rad, 1610747) and analysed by SDS-PAGE using Mini-PROTEAN TGX Gels 4% to 20% (Bio-Rad, 4561093). For the analysis of mitochondria-enriched fractions, 30 μg mitochondrial protein was aliquoted into a tube, centrifuged for 10 min at 16,000 × g, the supernatant removed and the pellet resuspended in Laemmli Sample Buffer (Bio-Rad, 1610747) to SDS-PAGE analysis as above. Gels were transferred onto pre-cut and soaked PVDF membranes (1704156, BioRad) using the BioRad Transblot Turbo transfer system, and blots were immediately stained with No-Stain Protein Labeling Reagent (Invitrogen, A44449) where indicated, according to the manufacturer's instructions. Fluorescence intensity was measured using a BioRad Chemidoc MP using the IR680 setting. Blots were then washed by gentle shaking 3 times for 5 min in PBS containing 0.1% (v/v) Tween-20 (PBST) and blocked by incubation with PBST containing 3% (w/v) BSA for 1 h. Blots were washed a further 3 times as above then incubated at 4˚C overnight with primary antibodies in PBST containing 3% (w/v) BSA. A further 3 washes were performed then the blots were incubated for 1 h in secondary antibodies made up in PBST containing 3% (w/v) BSA. Blots were then washed twice in PBST and once in PBS prior to incubation with ECL Prime western blotting system (Cytiva, RPN2232). Blots were imaged using the BioRad Chemidoc MP using exposure settings to minimise overexposure. Image analysis was performed using FIJI (Image J) and images were exported as TIFF files for publication. For paraquat treatment, flies were maintained in tubes (10 to 20 flies per replicate) containing 6 semi-circular pieces of filter paper (90 mm diameter, Cat#1001–090) saturated with 5% (w/v) sucrose solution containing 10 mM paraquat. Sucrose-only starvation experiments were performed as above, with the omission of paraquat. After 3 days, the flies were anaesthetised with mild $CO_2$ and live flies only were harvested and process for immunoblotting as described above.

## ARPE-19-MQC cells

ARPE-19-MQC-FIS1 cells were used to assess mitophagy in a human cell model. Briefly, cells were transfected in reverse with RNAiMax (13778075, Thermo Fisher Scientific) and non-targeting control siRNA (ON-TARGETplus Non-targeting Control Pool, D-001810-10-20, Horizon Discovery) or FBXO7 siRNA (ON-TARGETplus Human FBXO7 siRNA, L-013606-00-0010, Horizon Discovery). After 48 h, knockdown cells were seeded into and Ibidi dish (IB-81156) Thistle Scientific Ltd), and after 72 h cells were imaged live by a spinning disk microscope and generated images were processed using Imaris as previously described.

## Statistical analysis

Data are reported as mean ± SEM or mean ± 95% CI as indicated in figure legends, and the *n* numbers of distinct biological replicates are shown in each graph. For statistical analyses of

lifespan experiments, a log rank (Mantel–Cox) test was used. For behavioural analyses, a Kruskal–Wallis nonparametric test with Dunn's post hoc test correction for multiple comparisons was used. Where multiple groups were compared, statistical significance was calculated by one-way ANOVA with Bonferroni post hoc test correction for multiple samples. This test was applied to dopaminergic neuron analysis and mitophagy where more than 2 groups were compared. For Ub and pS65-Ub abundance, a one-way ANOVA with Dunnett's correction for multiple comparisons was used. When only 2 groups were compared, a Welch's *t* test was used. The absence of statistical analysis in between any group reflects "no statistical significance." All the samples were collected and processed simultaneously and therefore no randomisation was appropriate. Unless otherwise indicated, all the acquisition of images and analysis was done in blind conditions. Statistical analyses were performed using GraphPad Prism 9 software (GraphPad Prism, RRID:SCR_002798). Statistical significance was represented in all cases as follows: * $P < 0.05$, ** $P < 0.01$, *** $P < 0.001$, and **** $P < 0.0001$.

## Supporting information

**S1 Fig. *mito-QC* reporter targeted to the mitochondrial matrix (*mtx-QC*) reproduces the results observed with the *mito-QC*.** (**A**, **B**) Confocal microscopy analysis of the *mtx-QC* reporter in larval CNS of control, *ntc* mutant, *ntc* overexpression and USP30 knockdown with the pan-neuronal driver *nSyb*-GAL4. Mitolysosomes are evident as GFP-negative/mCherry-positive (red-only) puncta; *n* shown in chart. One-way ANOVA with Bonferroni post hoc test correction; * $P < 0.05$. Scale bars = 10 μm. Full details of numerical data and analyses underlying the quantitative data can be found in S1 Data.
(TIFF)

**S2 Fig. FBXO7 affects basal mitophagy in vivo *Drosophila* neurons and in a human cell line.** (**A**, **B**) Confocal microscopy analysis of the *mito-QC* reporter in larval CNS of control and the transgenic expression of *FBXO7* with the pan-neuronal driver *nSyb*-GAL4. Mitolysosomes are evident as GFP-negative/mCherry-positive (red-only) puncta; *n* shown in chart. One-way ANOVA with Bonferroni post hoc test correction; *** $P < 0.001$. Scale bar = 10 μm. (**C**) Immunoblot analysis of the knockdown of FBXO7 in human ARPE-19 cells expressing the *mito-QC* reporter. Arrow shows FBXO7 band; * shows nonspecific band. (**D**, **E**) Confocal microscopy analysis of the *mito-QC* reporter in ARPE-19 human cell line of control siRNA and FBXO7 siRNA. Mitolysosomes are evident as GFP-negative/mCherry-positive (red-only) puncta; *n* shown in chart. Two-tailed *t* test; ** $P < 0.01$. Scale bars = 10 μm. Full details of numerical data and analyses underlying the quantitative data can be found in S1 Data.
(TIFF)

**S3 Fig. USP30 but neither March5 nor MUL1 can modulate mitophagy in vivo.** (**A**, **B**) Confocal microscopy analysis of the *mito-QC* reporter in 2-day-old adult thoraces of control and USP30 knockdown with the muscular driver *Mef2*-GAL4. Mitolysosomes are evident as GFP-negative/mCherry-positive (red-only) puncta; *n* shown in chart. Two-tailed *t* test; * $P < 0.05$. Scale bar = 10 μm. (**C**) Climbing ability of 2-day-old flies expressing *USP30* RNAi with the ubiquitous driver *da*-GAL4 or with the pan-neuronal driver *nSyb*-GAL4. Chart show mean ± 95% CI and *n* values. Kruskal–Wallis nonparametric test with Dunn's post hoc test correction for multiple comparisons. (**D**) Climbing ability of 10-day-old flies overexpressing USP30 alone or in combination with parkin, March5 or Mul1 with the ubiquitous driver *Act*-GAL4. Chart show mean ± 95% CI and n values. Kruskal–Wallis nonparametric test with Dunn's post hoc test correction for multiple comparisons; **** $P < 0.0001$. (**E**, **F**) Confocal microscopy analysis of the *mito-QC* reporter in larval CNS of control and parkin

overexpression with the pan-neuronal driver *nSyb*-GAL4. Mitolysosomes are evident as GFP-negative/mCherry-positive (red-only) puncta; *n* shown in chart. Two-tailed *t* test. Scale bar = 10 μm. (**G, H**) Confocal microscopy analysis of the *mito-QC* reporter in larval CNS of control and *Mul1* mutant. Mitolysosomes are evident as GFP-negative/mCherry-positive (red-only) puncta; *n* shown in chart. Two-tailed *t* test. Scale bar = 10 μm. (**I, J**) Confocal microscopy analysis of the *mito-QC* reporter in larval CNS of control and March5 knockdowns with the pan-neuronal driver *nSyb*-GAL4. Mitolysosomes are evident as GFP-negative/mCherry-positive (red-only) puncta; *n* shown in chart. Two-tailed *t* test. Scale bar = 10 μm. Full details of numerical data and analyses underlying the quantitative data can be found in S1 Data.
(TIFF)

**S4 Fig. Increase in mitophagy levels requires the critical autophagy factor Atg8.** (**A, B**) Confocal microscopy analysis of the *mito-QC* reporter in larval CNS of control, knockdown of USP30, overexpression of ntc, and *ntc* mutant in the *Atg8a* mutant background with the pan-neuronal driver *nSyb*-GAL4. Mitolysosomes are evident as GFP-negative/mCherry-positive (red-only) puncta; *n* shown in chart. One-way ANOVA with Bonferroni post hoc test correction; * $P < 0.05$, *** $P < 0.001$. Scale bar = 10 μm. (**C, D**) Confocal microscopy analysis of the *mito-QC* reporter in larval CNS of control and *parkin* mutant alone or with the USP30 knockdown in the presence or absence of *Atg8a*, with the pan-neuronal driver *nSyb*-GAL4. Mitolysosomes are evident as GFP-negative/mCherry-positive (red-only) puncta; *n* shown in chart. One-way ANOVA with Bonferroni post hoc test correction; * $P < 0.05$, ** $P < 0.01$, **** $P < 0.0001$. Scale bar = 10 μm. Full details of numerical data and analyses underlying the quantitative data can be found in S1 Data.
(TIFF)

**S5 Fig. Analysis of total Ub in subcellular fractions and pS65-Ub upon oxidative stress-induction.** (**A**) Representative immunoblot of the subcellular fractionation of 2-day-old flies with the transgenic expression of *ntc* or *USP30* RNAi with the ubiquitous driver *da*-GAL4. Cytosolic- and mitochondria-enriched fractions are label with Actin and ATP5a, respectively. (**B**) Representative immunoblot of total ubiquitin (FK2) of 2-day-old flies in control, *ntc* overexpression and *USP30* knockdown with the ubiquitous driver *da*-GAL4. (**C**) Representative immunoblot of pS65-Ub levels of 2-day-old flies treated with paraquat (PQ) in control and *ntc* mutants.
(TIFF)

**S1 Table. Details of full genotypes used in this study. More details of each line can be found in Materials and methods.**
(DOCX)

**S1 Data. Full details of numerical data and analyses underlying the quantitative data.**
(XLSX)

**S1 Raw Images. Original scan images for Figs 1E, 1H, 6A, 6C, 7A, 7C, 7E, S2C, S5A, S5B, and S5C.**
(PDF)

## Acknowledgments

We kindly thank Herman Steller for generously sharing the ntc lines, Ugo Mayor for the USP30 overexpression lines, and Ian Ganley for the ARPE-19-MQC cells. We thank Roberta Tufi for performing the ATP assay, Wing Hei Au and Federica De Lazzari for comments and

edits on the manuscript, and all the members of the Whitworth's lab for discussions. Stocks were obtained from the Bloomington *Drosophila* Stock Center which is supported by grant NIH P40OD018537.

## Author Contributions

**Conceptualization:** Alvaro Sanchez-Martinez, Aitor Martinez, Alexander J. Whitworth.

**Data curation:** Alvaro Sanchez-Martinez.

**Formal analysis:** Alvaro Sanchez-Martinez.

**Funding acquisition:** Aitor Martinez, Alexander J. Whitworth.

**Investigation:** Alvaro Sanchez-Martinez, Aitor Martinez, Alexander J. Whitworth.

**Methodology:** Alvaro Sanchez-Martinez.

**Project administration:** Alvaro Sanchez-Martinez, Aitor Martinez, Alexander J. Whitworth.

**Supervision:** Alvaro Sanchez-Martinez, Aitor Martinez, Alexander J. Whitworth.

**Validation:** Alvaro Sanchez-Martinez, Aitor Martinez, Alexander J. Whitworth.

**Visualization:** Alvaro Sanchez-Martinez, Aitor Martinez, Alexander J. Whitworth.

**Writing – original draft:** Alvaro Sanchez-Martinez, Aitor Martinez, Alexander J. Whitworth.

**Writing – review & editing:** Alvaro Sanchez-Martinez, Aitor Martinez, Alexander J. Whitworth.

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
