## [Editor Report · Decision Letter 0]

23 Apr 2023

Dear Dr Sanchez-Martinez, 

Thank you for submitting via Review Commons your revised manuscript entitled "FBXO7/ntc and USP30 antagonistically set the ubiquitination threshold for basal mitophagy and provides a target for Pink1 phosphorylation in vivo" for consideration as a Research Article by PLOS Biology.

Your manuscript has now been evaluated by the PLOS Biology editorial staff as well as by an academic editor with relevant expertise and I am writing to let you know that we would like to send your submission back to the original reviewers.

Once your full submission is complete, your paper will undergo a series of checks in preparation for peer review. After your manuscript has passed the checks it will be sent out for review. To provide the metadata for your submission, please Login to Editorial Manager (https://www.editorialmanager.com/pbiology) within two working days, i.e. by Apr 25 2023 11:59PM.

Kind regards,

Ines

--

Ines Alvarez-Garcia, PhD

Senior Editor

PLOS Biology

---

## [Decision Letter · Decision Letter 1]

30 May 2023

Dear Dr Sanchez-Martinez,

Thank you for your patience while we considered your revised manuscript entitled "FBXO7/ntc and USP30 antagonistically set the ubiquitination threshold for basal mitophagy and provides a target for Pink1 phosphorylation in vivo" for publication as a Research Article at PLOS Biology. This revised version of your manuscript has been evaluated by the PLOS Biology editors, the Academic Editor and the three original reviewers from Review Commons.

Based on the reviews, we are likely to accept this manuscript for publication, provided you satisfactorily address the policy-related requests stated below.

In addition, we would like you to consider a suggestion to improve the title:

"The ubiquitination threshold for basal mitophagy is set antagonistically by USP30 and the Pink1 target FBXO7/ntc"

We expect to receive your revised manuscript within two weeks. 

*Published Peer Review History*

*Press*

Sincerely,

Ines

--

Ines Alvarez-Garcia, PhD

Senior Editor

PLOS Biology

Fig. 1A, B, F, G; Fig. 2A-D, G-I; Fig. 3B, D, F, G; Fig. 4B, D; Fig. 5B, D; Fig. 6B, D, F; Fig. 7B, D, F; Fig. S1B; Fig. S2B, E; Fig. S3B-D, H, J and Fig. S4B, D

We require the original, uncropped and minimally adjusted images supporting all blot and gel results reported in an article's figures or Supporting Information files. We will require these files before a manuscript can be accepted so please prepare and upload them now. Please carefully read our guidelines for how to prepare and upload this data: https://journals.plos.org/plosbiology/s/figures#loc-blot-and-gel-reporting-requirements

BLURB

Please also provide a blurb which (if accepted) will be included in our weekly and monthly Electronic Table of Contents, sent out to readers of PLOS Biology, and may be used to promote your article in social media. The blurb should be about 30-40 words long and is subject to editorial changes. It should, without exaggeration, entice people to read your manuscript. It should not be redundant with the title and should not contain acronyms or abbreviations. For examples, view our author guidelines: https://journals.plos.org/plosbiology/s/revising-your-manuscript#loc-blurb

Reviewers' comments:

Rev. 1:

The authors adequately addressed my critiques and the manuscript can be published in the current form.

Rev. 2:

Recommended for publication. Thank you to the authors for addressing my queries.

Rev. 3:

Accept.

---

## [Editor Report · Decision Letter 2]

11 Jul 2023

Dear Dr Sanchez-Martinez,

Thank you for the submission of your revised Research Article entitled "FBXO7/ntc and USP30 antagonistically set the ubiquitination threshold for basal mitophagy and provide a target for Pink1 phosphorylation in vivo" for publication in PLOS Biology. On behalf of my colleagues and the Academic Editor, Yi-Hsien Su, I am delighted to let you know that we can in principle accept your manuscript for publication, provided you address any remaining formatting and reporting issues. These will be detailed in an email you should receive within 2-3 business days from our colleagues in the journal operations team; no action is required from you until then. Please note that we will not be able to formally accept your manuscript and schedule it for publication until you have completed any requested changes.

PRESS

Sincerely, 

Ines

--

Ines Alvarez-Garcia, PhD

Senior Editor

PLOS Biology
